# Quantification of Glucose Metabolism and Nitrogen Utilization in Two *Brassicaceae* Species under Bicarbonate and Variable Ammonium Soil Conditions

**DOI:** 10.3390/plants12173095

**Published:** 2023-08-29

**Authors:** Antong Xia, Yanyou Wu, Jiqian Xiang, Hongqing Yin, Jiajia Ming, Zhanghui Qin

**Affiliations:** 1Enshi Tujia & Miao Autonomous Prefecture Academy of Agricultural Sciences, Enshi 445000, China; tone1214910327@163.com (A.X.);; 2State Key Laboratory of Environmental Geochemistry, Institute of Geochemistry, Chinese Academy of Sciences, Guiyang 550081, China

**Keywords:** EMP, PPP, NO_3_^−^, NH_4_^+^, biodirectional isotope tracing technique, karst adaption

## Abstract

In karst habitats under drought conditions, high bicarbonate (high pH), and an abundant nitrate soil environment, bicarbonate regulates the glycolysis (EMP) and pentose phosphate pathways (PPP), which distribute ATP and NADPH, affecting nitrate (NO_3_^−^) and ammonium (NH_4_^+^) utilization in plants. However, the relationship between EMP PPP and NO_3_^−^, and NH_4_^+^ utilization and their responses to bicarbonate and variable ammonium still remains elusive. In this study, we used *Brassica napus* (*Bn*, a non-karst-adaptable plant) and *Orychophragmus violaceus* (*Ov*, a karst-adaptable plant) as plant materials, employed a bidirectional nitrogen-isotope-tracing method, and performed the quantification of the contribution of EMP and PPP. We found that bicarbonate and ammonium inhibited glucose metabolism and nitrogen utilization in *Bn* under simulated karst habitats. On the other hand, it resulted in a shift from EMP to PPP to promote ammonium utilization in *Ov* under high ammonium stress in karst habitats. Compared with *Bn*, bicarbonate promoted glucose metabolism and nitrogen utilization in *Ov* at low ammonium levels, leading to an increase in photosynthesis, the PPP, carbon and nitrogen metabolizing enzyme activities, nitrate/ammonium utilization, and total inorganic nitrogen assimilation capacity. Moreover, bicarbonate significantly reduced the growth inhibition of *Ov* by high ammonium, resulting in an improved PPP, RC_RUBP_, and ammonium utilization to maintain growth. Quantifying the relationships between EMP, PPP, NO_3_^−^, and NH_4_^+^ utilization can aid the accurate analysis of carbon and nitrogen use efficiency changes in plant species. Therefore, it provides a new prospect to optimize the nitrate/ammonium utilization in plants and further reveals the differential responses of inorganic carbon and nitrogen (C-N) metabolism to bicarbonate and variable ammonium in karst habitats.

## 1. Introduction

In high bicarbonate (high pH) karst ecosystems, under drought, bicarbonate (HCO_3_^−^) is the main product of carbonate after karstification [1]. Previous studies have shown that high bicarbonate stress causes plant growth inhibition due to a decline in photosynthesis [2], nitrogen use efficiency [3], and carbon and nitrogen metabolism enzyme activities [4,5]. Recently, many scholars have found that HCO_3_^−^ could promote plants’ growth under drought, high bicarbonate, and high pH stress in karst environments by enhancing the bicarbonate use capacity (BUC) and total inorganic carbon assimilation capacity of *Camptotheca acuminata Decne* [6] and *Broussonetia papyrifera* [7]. Moreover, bicarbonate photolysis and water photolysis could both be sources of photosynthetic oxygen [8,9]. This has resulted in a revised formula of photosynthetic oxygen evolution as follows: 2H_2_O + CO_2_ → H_2_O + H^+^ + HCO_3_^−^ → O_2_ + 4e^−^ + 4H^+^ + CO_2_ [10], which should be sourced from both bicarbonate and water in a 1:1 (mol/mol) stoichiometric relationship [11]. Hence, it confirmed that bicarbonate significantly contributes to plant growth in karst habitats.

Additionally, bicarbonate could positively regulate glucose metabolism in plants, primarily through the glycolysis pathway (EMP) and pentose phosphate pathway (PPP), to facilitate adaptation to karst habitats [5,12]. Wu and Xing [13] determined the contribution of PFK and G6PDH (key enzymes in EMP and PPP, respectively) and total glucose metabolism to determine the adaptation of plant species to karst environments. Therefore, the photosynthetic growth power (GC) and RUBP regeneration capacity (RC_RUBP_) could also be calculated to estimate the capacity of glucose (organic carbon) disproportionation from photosynthesis to the EMP and PPP. High bicarbonate stimulates the shift from the EMP to the PPP, and in the case of *Broussonetia papyrifera (Bp.*), both the EMP and PPP are increased at 3 mM HCO_3_^−^ [12]. Notably, 10 mM HCO_3_^−^ promoted the pentose phosphate pathway of *Orychophragmus violaceus* (*Ov*) while maintaining its growth [5]. 

Bicarbonate also enhanced nitrate use in plant species grown in karst environments [14,15]. Nitrogen assimilation uses both ATP and NADPH from the EMP and PPP, which impacts nitrate reduction and ammonium assimilation efficiency. Nitrate and ammonium are commonly used in inorganic nitrogen sources of plant tissue culture and utilized in complex processes of nitrogen metabolism, including nitrate reduction, ammonium assimilation, and protein formation [16]. Therefore, the unidirectional isotope method can be implemented to precisely calculate nitrate and ammonium utilization, albeit with some difficulties. Generally, it is difficult to directly obtain plants’ nitrate/ammonium utilization by traditional methods. Fortunately, the bidirectional stable nitrogen isotope tracers can accurately quantify the nitrate and ammonium utilization by plants [13,17]. Under the same culture conditions, we used two nitrogen isotope markers with a difference in *δ*^15^N value over 10‰, labeled with high (H) and low (L), to obtain nitrate utilization/ammonium by different plant species accurately. In addition, according to the NR and GOGAT activities, nitrate/ammonium assimilation efficiency in plant species can be determined [18,19]. 

Due to drought, high pH, and high bicarbonate in the karst soil, the NH4+ is easily volatilized to NH3 and enters the atmosphere. Consequently, the content of ammonium is often lower, while nitrate in karst soil is often higher than that in non-karst areas [13]. Soils high in nitrate and bare in ammonium are typical characteristics under karst areas, with the lack of ammonium being a key factor limiting plant growth [20]. The assimilation of one NO_3_^−^ consumes 20 mol of ATP, while the assimilation of one NH_4_^+^ only consumes 5 mol of ATP. Consequently, the energy cost for the assimilation of 1 mol of nitrogen (NO_3_^−^ + NH_4_^+^ = 1 mol total nitrogen) will be in the range of 5 mol to 20 mol of ATP for plant species’ grown in mixed nitrogen source conditions [21], which is much higher compared to exclusively ammonium assimilation (only 5 ATP/mol NH_4_^+^). However, high ammonium can lead to the acidification of the root, inhibiting the nitrogen accumulation efficiency (NAC) and plant growth [22]. It is not clear whether there is a linear relationship between ammonium supply (total nitrogen concentration) and ammonium utilization among different plant species. Furthermore, it has been reported that the EMP and PPP pathways are two important metabolic processes in glucose metabolism, involved in the production of ATP and NAD(P)H, respectively. The contributions of ATP and NAD(P)H were determined when plant species assimilated 1 mol of glucose [12,13]. Therefore, based on the EMP and PPP contributions, we can predict the production efficiency of ATP and NAD(P)H in plant species. However, the relationship between EMP, PPP, NO_3_^−^, and NH_4_^+^ utilization in plant species and their differential responses to bicarbonate and variable ammonium have not been directly investigated. 

In this study, we experimentally evaluate the growth of *Brassica napus* (*Bn*, a non-karst-adaptable plant) and *Orychophragmus violaceus* (*Ov*, a karst-adaptable plant), which are both *Brassicaceae* species. *Bn* is an important cash crop widely grown for oil production, and *Ov* is cultivated due to its fatty acid content and fuel properties [23]. Bicarbonate concentrations in karst soils are usually in the range of 1–10 mM; thus, 10 mM NaHCO_3_ was used to simulate a high bicarbonate (high pH) environment with a relatively stable pH, set at 8.3 ± 0.02 [24]. It should be emphasized that Na^+^ presence merely inhibited plant growth because 10 mM Na^+^ is much lower than the toxic concentration of Na^+^ in plants (more than 50 mM Na^+^) [25]. Hence, we aim to answer the following questions: (1) based on the bidirectional N stable isotope tracing method, we aim to reveal the different utilizations of NO_3_^−^ and NH_4_^+^ in bicarbonate and various ammonium conditions; and (2) to precisely quantify the relationship between EMP, PPP, and NO_3_^−^/NH_4_^+^ utilization capacity in a non-karst-adaptable plant and a karst-adaptable plant.

## 2. Materials and Methods

### 2.1. Bicarbonate and Ammonium Treatment

Two Brassicaceae plants, *Bn* (a non-karst-adaptable plant) and *Ov* (a karst-adaptable plant), were selected as the experimental materials. The *Bn* and *Ov* seeds were obtained from the Guizhou Vocational College of Agriculture, Guizhou, and Shanxi Agricultural Reclamation Scientific Research Center, Shanxi, China. The experiments were conducted in a greenhouse with a length, width, and height of 10 × 5 × 4 m (respectively) at the Institute of Geochemistry, Chinese Academy of Sciences (Guiyang, China). Metal halide lamps (HPI-T400 W/645, Philips, The Netherlands) were used as a light source, and the temperature was controlled by air conditioning. The greenhouse environment was maintained as follows: light 500 ± 23 μmol m^−2^·s^−1^; temperature (day/night) of 25/19 °C; constant light time of 12 h per day; and relative humidity range of 55–60%. The *Bn* and *Ov* seeds were stirred uniformly with 70% ethanol, sterilized for 1 min, repeatedly rinsed with sterile water 3–5 times, and soaked for 6–8 h. The *Ov* and *Bn* seedlings were grown in germination trays (twelve-hole size, 19 × 15 × 9.5 cm), filled with perlite: vermiculite in a 1:3 ratio, and irrigated with a modified Hoagland solution that provided the nutrients. In addition, it has been reported that karst drought is characterized by high pH and high bicarbonate. Therefore, we applied PEG6000 with at a 10 g/L concentration to simulate a drought habitat, and the pH was maintained at 8.30 ± 0.02. The nutrient solution was changed every 3 days, and the seedlings were transplanted after 28 days for the subsequent experiment. The *Bn* and *Ov* seedlings with uniform growth were categorized as 3 plants/pot, 3 pots/group, and 3 groups/treatment for the differential bicarbonate treatments, which were conducted for 10 days. A total of 10 mM NaHCO_3_ was added to simulate a high pH (8.30 ± 0.05), referring to the NO_3_^−^ concentration from the Hoagland nutrient solution. A concentration of 15 mM NO_3_^−^ was used in this experiment. Meanwhile, based on the bidirectional stable nitrogen isotope-tracing method, we selected two groups of ^15^NO_3_^−^ with isotopic values differing by more than 10‰, which were labeled as High (H) and Low (L) to culture *Bn* and *Ov* at different NH_4_^+^ levels (Table 1). After 8 days, the following indicators were measured. 

### 2.2. Sample Collection for Biomass Estimation

The *Bn* and *Ov* plants, under various treatments, were collected in three parts: roots, stem, and leaves. They were heated at 108 °C for 30 min and dried at 70 °C to obtain the organic biomass and total biomass.

### 2.3. Photosynthesis Measurement

The 2nd to 3rd fully expanded leaves of *Ov* and *Bn* plants were used to measure photosynthesis from 9:00 to 11:00 a.m. The Li-6400 photosynthesis system (LI-COR, Lincoln, Raleigh, USA) was used to measure photosynthesis, including the net photosynthetic rate (Pn, μmol m^−2^·s^−1^), stomatal conductance (Gs, mmol H_2_O m^−2^·s^−1^), transpiration rate (Tr, mmol H_2_O m^−2^·s^−1^), and intercellular CO_2_ concentration (Ci, μmol CO_2_ mol^−1^·air^−1^) in *Bn* and *Ov* plants. The water use efficiency (WUE) was analyzed according to Formula (1).
WUE = Pn/Tr(1)

The other parameters were set as follows: temperature of 25 °C, CO_2_ concentration of 400 μmol/mol in the buffered glass bottles, and photosynthetically active radiation intensity of 500 μmol/m^2^·s^−1^.

### 2.4. C/N Concentration of Leaves

Dried leaves from *Bn* and *Ov* plants were used to determine the carbon and nitrogen contents using an elemental analyzer (Vario MACRO cube, Frankfurt, Germany). The C and N concentrations of leaves were measured according to Formulas (2) and (3), respectively.
C (leaves, mg/g) = Dwleaves × C%(2)
N (leaves, mg/g) = DWleaves × N%(3)

### 2.5. Carbon and Nitrogen Metabolism Enzymes 

Fresh leaves (0.1 g) from *Bn* and *Ov* plants (the 3rd or 4th leaves of the seedlings) were weighed and ground with liquid nitrogen. Then, the Rubisco (ribulose bisphosphate carboxylase oxygenase, A_Rubisco_), SS (sucrose synthase, A_SS_), NR (nitrate reductase, A_NR_), GOGAT (Glutamate synthase, A_GOGAT_), PFK (Phosphofructokinase, A_PFK_), and G6PDH (NADP-glucose-6-phosphate dehydrogenase, A_G6PDH_) activities were measured with bio enzyme kits (Sangon Biotech, Shanghai, China). Carbon- and nitrogen-metabolizing enzymes were measured as follows:

A_Rubisco_: In 0.1 g of tissue, 1 mL of extract A was added and was homogenized in an ice bath by sonication (200 W, sonication 3 s, interval 7 s, 1 min). Then, the solution was centrifuged at 4 °C, 8500 rpm for 10 min. The supernatant was taken as the crude enzyme solution and placed on ice for subsequent measurements. The measurements were performed in a UV spectrophotometer, which was preheated for more than 30 min; the wavelength was set to 340 nm, and distilled water was used as a blank. According to the instruction of the Rubisco kit, the reagents numbered 1–5 were added in a 1 mL quartz cuvette at 25 °C, mixed gently, and the Rubisco activity was measured at 340 nmOD, reading the absorption values A1 and A2 after 10 s and 10 min (ΔA = Arubisco = A1–A2). The Rubisco activity was then calculated according to the kit (Order No. D799834-9100, Sangon Biotech).

A_SS_: In 2.0 g of plant sample, 2 mL of acetic acid buffer was added, and the mix was ground into a paste with a mortar in an ice bath. It was then centrifuged at 12,000 r/min for 10 min, and the supernatant was taken for enzyme activity determination. A total of 0.8 mL of the sample, 0.5 mmol/L acetic acid buffer, 0.2 mL sucrose solution, and 1 mL diluted enzyme solution were added into two stoppered, graduated test tubes and left at room temperature for 10 min. The absorbance was measured at 25 °C, pH = 4.5, and OD510 nm. A1 and A2 were obtained after 10 s and 1 min (ΔA = ASS = A1–A2), and then the Rubisco activity was calculated according to the SS kit (Order No. D799786, Sangon Biotech).

A_NR_: In 0.1 g of plant sample, 1 mL of extraction solution was added, and the mix was homogenized in an ice bath (or ground with quartz sand). It was then centrifuged at 12,000 rpm for 10 min at 4 °C. The supernatant was removed and placed on ice for measurement. The visible spectrophotometer was preheated for more than 30 min, the wavelength was adjusted to 340 nm, and distilled water was used as the blank. The sample (80 μL), reagent 1 (280 μL), and reagent 2 (120 μL) were added into the EP tube in turn, and the reaction was conducted at 30 °C for 15 min. All the clear liquid was transferred to a 1 mL glass cuvette. The absorbance value A was obtained immediately at 530 nm, with ΔA = ANR = Ameasurement-Acontrol, and the NR activity was then obtained according to the instructions of the NR kit (Order NO. D799304, Sangon Biotech).

A_GOGAT_: In 0.1 g of plant sample, 1 mL of extraction solution was added, and the mix was homogenized in an ice bath. It was then centrifuged at 8000× *g* for 10 min at 4 °C, and the supernatant was taken and placed on ice for measurement. The spectrophotometer was preheated for more than 30 min, the wavelength was adjusted to 340 nm, and distilled water was used as the blank. A total of 25 mL of reagent 1 was added to reagent 2, dissolved and mixed thoroughly, and placed in a water bath at 25 °C for 5 min. A total of 0.1 mL of sample and 0.9 mL of working solution was placed in a 1 mL cuvette and mixed well. The initial absorbance A1 was recorded at 20 s, and the absorbance A2 at 320 s at 340 nm wavelength. ΔA = A_GOGAT_ = A1–A2 was calculated, and then the GOGAT value of the sample was obtained according to the instructions of the kit (Order NO. D799302, Sangon Biotech). 

A_PFK_: For the phosphofructokinase (PFK) activity assay, ATP-PFK and PPi-PFK were assayed as described previously with some modifications. A 200μL aliquot of the extract was added to 1.8 mL of the assay buffer containing 50 mM Hepes-Tris (pH 7.8), 2.5 mM MgCl_2_, 0.1 mM NADH, 5 mM F-6-P, 2 units/mL aldolase, 1 unit/mL triosephosphate isomerase, 2 units/mLα-glycerol-3-phosphate dehydrogenase, and either 1 mM ATP or 1 mM PPi. The oxidation of NADH to NAD^+^ was measured as the changing rate of the absorbance at 340 nm during the initial 5 min. The PFK activity was equal to ATP-PFK activity and PPi-PFK activity.

A_G6PDH_: The Glucose-6-phosphate dehydrogenase (G6PDH) activity was assayed as described previously with some modifications. A 200 μL aliquot of the extract was added to either 1.8 mL of the total dehydrogenase (G6PDH + 6PGDH) assay buffer containing 50 mM Hepes-Tris (pH 7.8), 3.3 mM MgCl_2_, 0.5 mM D-glucose-6-phosphate disodium salt, 0.5 mM 6-phosphogluconate, and 0.5 mM NADP-Na2 or 1.8 mL, and the 6-phosphogluconate activity was subtracted.

### 2.6. Evaluation of the Glycolysis Pathway and the Pentose Phosphate Pathway Activities

The total glucose metabolic activity (EA_∑_), the contribution of EMP (E_EMP_) and of PPP(E_PPP_), the growth capacity (GC), and the regeneration capacity of Ribulose-1,5-bisphosphate (RC_RUBP_) were calculated according to Formulas (4)–(8), respectively [12].
EA_∑_ = A_PFK_ + A_G6PDH_(4)
E_EMP_ = A_PFK_/EA_∑_(5)
E_PPP_ = A_G6PDH_/EA_∑_(6)
GC = E_EMP_ × Pn(7)
RC_RUBP_ = E_PPP_ × Pn(8)

### 2.7. Measurement of δ^15^ N Values 

The *δ*^15^N value of plant leaves was measured by a gas isotope ratio mass spectrometer (MAT 253, Germany) according to Formula (9):*δ*^15^ N sample = R_sample_/R_standard_ − 1 × 1000(9)

### 2.8. Utilization of NO_3_^−^/NH_4_^+^ in Plants

Mixed nitrogen sources, including nitrate and ammonium, have been used to determine the nitrogen isotope values in plants. In this study, two nitrate sources with different nitrogen isotope values (more than 10‰) were supplied for nitrate utilization. The *δ*^15^N value of ammonium was lower compared to both nitrate sources. Therefore, the *δ*^15^N value reflected the results of assimilated nitrate and ammonium. Consequently, using Formula (10), we calculated the foliar *δ*^15^N value (*δ*_T_), obtained by measuring the nitrogen isotope composite ion. Subsequently, the bidirectional nitrogen isotope method was calibrated to quantify the contributions of nitrate and ammonium. *δ*_A_ and *δ*_B_ are the *δ*^15^N values derived from nitrate/ammonium assimilation, and *f*_A_ and *f*_B_ are the contributions of NO_3_^−^/NH_4_^+^ assimilation in plants.
*δ*_T_ = *f*_A_ *δ*_A_ + *f*_B_ *δ*_B_ = *f*_A_
*δ*_A_ + (1 − *f*_A_) *δ*_B_(10)

In this study, two labeled nitrogen isotope ratios, which differed by 10% (*δ*^15^N values, L: 12.7‰; H: 22.72‰), were used to quantify the nitrate (*f*_A_) and ammonium contribution (*f*_B_). The bidirectional nitrogen isotope tracing method was calculated as follows (11).
*δ*_TH_ = *f*_AH_*δ*_AH_ + *f*_B_*δ*_B_ = *f*_AH_*δ*_AH_ +(1 − *f*_AH_) *δ*_B_(11)

In contrast, the low isotope treatment measurement can be expressed as follows (12).
*δ*_TL_ = *f*_AL_*δ*_AL_+ *f*_B_*δ*_B_ = *f*_AL_*δ*_AL_ + 1 − *f*_AL_ − *δ*_B_(12)

After treatment, the total nitrogen (TN) in plant leaves corresponds to the sum of the pre-existing nitrogen (before treatment) and the newly acquired nitrogen (at treatment). Therefore, the plant leaf nitrogen isotope values (*δ*_T_) are mixed *δ*^15^N values and are the sum of the *δ*^15^N values of the leaves before treatment and at treatment. In that case, we set the percentage of the leaf nitrogen before treatment as *f*_0_, the leaf *δ*^15^N values as *δ*_0_, the percentage of the leaf nitrogen post-treatment as fnew, and the leaf *δ*^15^N values as *δ*_new_. In conclusion, we obtained the different nitrogen isotope values of leaves, as shown in Formula (13).
*δ*_T_= *f*_0_*δ*_0_ + *f*_new_ *δ*_new_(13)

The sum of *f*_0_ and *f*_new_ is 1 (100%). Therefore, we can use Formula (14):*f*_new_ = 1 − *f*_0_(14)

Thus, Formula (13) can be expressed as Formula (15):*δ*_T_ = *f*_0_
*δ*_0_ + (1 − *f*_0_) *δ*_new_(15)

In addition, we could obtain *f*_0_ and *δ*_new_, which were set as in Formula (16):(16)$$f_{0}=\frac{\delta_{T}-\delta_{new}}{\delta_{0}-\delta_{new}}$$

In Formula (16), $\delta_{T}$ is the mixed nitrogen isotope value of the leaves; $\delta_{0}$ is the leaf nitrogen isotope value before treatment; and $\delta_{new}$ is the leaf nitrogen isotope value post-treatment. Initially, the value we found was 0, $\delta_{T}$ = $\delta_{0}$. Consequently, *f*_0_ was 1 (100%) at 0 days before treatment. In this experiment, *δ*_new_ corresponds to the *δ*_TL_ values of the nitrogen isotope markers, and the *δ*^15^N values of the high/low nitrogen isotope markers were 22.72‰ ($\delta_{new-H}$) and 12.7‰ ($\delta_{new-L}$), respectively. Hence, we obtained the different nitrogen isotope values of the leaves (*δ*_TH_ and *δ*_TL_) labeled with high/low nitrogen isotopes markers, as shown in Formulas (17) and (18):*δ*_TH_ = *f*_0_*δ*_0_ + (1 − *f*_0_) δ_new-H_(17)
*δ*_TL_ = *f*_0_*δ*_0_ + (1 − *f*_0_) δ_new-L_(18)

In this experiment, the same culture conditions were maintained for both plant species. Therefore, the nitrate and ammonium contributions were the same, except for the nitrogen isotope value in the high (H) and low (L) treatments. However, the physiological processes, metabolism, and growth were considered similar for both plants under the same total nitrogen level. Consequently, we obtained Formulas (19) and (20).
*f*_A_ = *f*_AH_ = *f*_A_(19)
1 − *f*_AH_ =1 − *f*_AL_(20)

Finally, the utilization of NO_3_^−^ (*f*_A_) and NH_4_^+^ (*f*_B_) in the plant species was calculated from Formulas (21) and (22), respectively:(21)$$f_{A}=\frac{\delta_{TH}-\delta_{TL}}{\delta_{H}-\delta_{L}}$$
*f*_B_ =1 (100%) − *f*_A_(22)

### 2.9. Total Nitrogen Assimilation Capacity of Plants

According to the *δ*^15^N values, the nitrogen assimilation capacity was expressed as Δ^15^N in Formula (23) [15], and the nitrogen accumulation capacity was estimated according to Formula (24).
Δ^15^N = *δ*^15^ N_substrate_ − *δ*^15^N_product_, *δ*^15^N_product_ = 8.08‰(23)
NAC (mg N/_plant_) = DW_plant_ × N% (24)

### 2.10. The Contribution of NO_3_^−^/NH_4_^+^ to Nitrogen Accumulation Capacity

The contribution of NO_3_^−^/NH_4_^+^ to total inorganic nitrogen accumulation (NACA/NACB) was calculated according to Formulas (25) and (26), respectively.
NACA = NAC × *f*_A_(25)
NACB = NAC × *f*_B_(26)

### 2.11. The ATP Consumption of NO_3_^−^, NH_4_^+^, and Total Nitrogen Assimilation

It has been reported that plants reduce 1 mol NO_3_^−^ consuming 20 mol ATP, while 1 mol NH_4_^+^ is assimilated by consuming 5 mol ATP [21]. Therefore, based on the NO_3_^−^ and NH_4_^+^ utilization values of the leaves (Formulas (21) and (22)), we obtained the ATP consumption for the nitrate reduction (ATPA), the ammonium assimilation (ATPB), and total nitrogen assimilation (ATP (A + B)) of leaves, as shown in Formulas (27)–(29), respectively.
ATPA = ATP (NO_3_^−^) × *f*_A_(27)
ATPB = ATP (NH_4_^+^) × *f*_B_
(28)
ATP (A + B) = ATPA + ATPB(29)

### 2.12. Statistical Analysis

The experimental data were evaluated for statistical differences using analysis of variance (ANOVA), and Tukey’s test (*p* ≤ 0.05) was performed for the pair-wise comparisons between the different experimental treatments. The results are presented as the mean ± standard deviation (SD), and the figures were designed using Origin Pro 2019b(64-bit).

## 3. Results

### 3.1. Plant Growth 

Significant differences were observed between *Bn* and *Ov* at different bicarbonate and ammonium levels (Table 2). In B0 (0 mM HCO_3_^−^), with increasing NH_4_^+^, the root, stem, leaf, and total biomass in *Bn* and *Ov* plants increased to the highest level at 10 mM NH_4_^+^ and 2 mM NH_4_^+^, respectively. On the other hand, in B10 (10 mM HCO_3_^−^), the biomass of root, stem, leaf, and total biomass in *Bn* was significantly lower than those in B0. Compared to *Bn*, the biomass of root, stem, leaf, and total biomass in *Ov* increased significantly at 2 mM NH_4_^+^ in B10.

### 3.2. Effects of Bicarbonate and Ammonium on Photosynthesis

In this study, the photosynthesis of *Bn* and *Ov* plants was remarkably contrasting at different bicarbonate and ammonium levels (Figure 1). With increasing NH_4_^+^, in B0, the Pn (net photosynthetic rate), Cs (intercellular carbon dioxide concentration), and Ci (intercellular carbon dioxide concentration) of *Bn* and *Ov* plants increased, reaching their maximum levels at 10 mM NH_4_^+^ and 2 mM NH_4_^+^, respectively. The WUE (water use efficiency) decreased to the lowest level at 20 mM NH_4_^+^.

With the addition of 10 mM HCO_3_^−^, the Pn, Gs, Ci, and WUE of *Bn* plants decreased significantly compared to the B0 treatment. However, the Pn of *Ov* plants did not change significantly, while the Gs and Tr (transpiration) significantly increased at 10 mM NH_4_^+^ and 20 mM NH_4_^+^.

### 3.3. Inorganic Carbon and Nitrogen Contents in Leaves

In Figure 2, under bicarbonate and ammonium treatments, the C, N, and NAC (nitrogen accumulation capacity) of *Bn* and *Ov* plants increased to the highest level at 2 mM NH_4_^+^ in B0, although it did not have a significant difference. On the other hand, significant differences were observed in B10. Compared to *Bn*, the C content of *Ov* plants increased significantly at 0.5 mM NH_4_^+^, and both the N content and NAC increased significantly at 2 mM NH_4_^+^. In contrast, they decreased consistently in *Bn* plants.

### 3.4. Carbon and Nitrogen Metabolism Enzymes 

The Rubisco, SS, NR, and GOGAT activities were measured in *Bn* and *Ov* plants (Figure 3). In B0, the Rubisco and SS activities of *Bn* and *Ov* plants were significantly increased to the highest level at 10 mM NH_4_^+^ and 2 mM NH_4_^+^, respectively. The NR activities of *Bn* and *Ov* plants reached their highest and lowest levels at 2 mM NH_4_^+^ and 20 mM NH_4_^+^, respectively, and significant differences were observed. The GOGAT activities of *Bn* and *Ov* plants reached their highest levels at 10 mM NH_4_^+^ and decreased at the highest ammonium level (20 mM NH_4_^+^).

In the B10 treatment, the Rubisco and SS activities of *Bn* plants significantly decreased, reaching the lowest level at 20 mM NH_4_^+^. The NR activities significantly decreased above 0.5 mM NH_4_^+^, while the GOGAT activities significantly decreased above 10 mM NH_4_^+^. Compared to *Bn*, the Rubisco activity in the *Ov* plants exhibited an overall increase at 2 mM NH_4_^+^ and the SS activity increased from 0.5 mM to 10 mM NH_4_^+^. In contrast, the NR and GOGAT activities of *Ov* plants decreased, reaching their lowest level at 20 mM NH_4_^+^.

### 3.5. Effects of Bicarbonate and Ammonium on the Glycolysis and Pentose Phosphate Pathways 

In Figure 4 of the B0 treatment, the PFK activity, E_EMP_, and total glucose metabolism increased, reaching their highest levels at 2 mM NH_4_^+^ in both *Bn* and *Ov*, while the G6PDH activities and the E_PPP_ of *Bn* and *Ov* plants were significantly increased only at 10 mM and 20 mM NH_4_^+^, respectively.

In the B10 treatment, the PFK activities and EMP of *Bn* plants decreased, reaching their lowest value at the highest ammonium level (20 mM NH_4_^+^). At this level, the G6PDH activities and PPP of *Bn* plants showed a contrasting response, increasing to their highest values. Therefore, the total glucose metabolism in *Bn* plants did not change compared to the B0 treatment. On the other hand, the PFK activities and EMP decreased, while the G6PDH activities, PPP, and total glucose metabolism increased in *Ov* plants.

### 3.6. The Growth Capacity and the Regeneration Capacity of RUBP

As shown in Figure 5, in the B0 treatment, the growth capacity (GC) and the regeneration capacity of RUBP (RC_RUBP_) of *Bn* and *Ov* plants reached their highest value at a low ammonium level (2 mM NH_4_^+^). In the B10 treatment, the GC and RC_RUBP_ of the *Bn* plants significantly decreased compared to B0, but the GC did not change, and the RC_RUBP_ increased at 2 mM NH_4_^+^ in *Ov* plants.

### 3.7. The Post-Treatment Nitrogen Isotope Values in the Leaves

As shown in Table 3, in the B0 treatment, the *δ*^15^N value in the *Bn* and *Ov* leaves significantly increased, reaching its highest level at 20 mM NH_4_^+^ and decreasing to its lowest level at 10 mM NH_4_^+^. The Δ^15^N values showed the opposite trend compared to the *δ*^15^N value, significantly decreasing to their lowest level at 20 mM NH_4_^+^ and increasing to their highest level at 10 mM NH_4_^+^.

In the B0 treatment, the *δ*^15^N value of *Bn* and *Ov* plants significantly increased at 0 mM and 10 mM NH_4_^+^. However, the *δ*^15^N value of *Bn* plants significantly decreased at the highest ammonium concentration (20 mM NH_4_^+^) in the B10 treatment, while the Δ^15^N value increased. Compared to *Bn*, the *δ*^15^N value of the *Ov* plants of B10 was significantly higher at the highest ammonium level (20 mM NH_4_^+^).

### 3.8. The NO_3_^−^/NH_4_^+^ Utilization of Leaves at Different Bicarbonate and Ammonium Treatments

In Table 4, at different bicarbonate and ammonium treatments, the utilization of NO_3_^−^ (*f*_A_) and NH_4_^+^ *(f*_B_) and their contributions to total inorganic nitrogen accumulation in *Bn* and *Ov* leaves reached their lowest levels at 10 mM NH_4_^+^, while *f*_B_ exhibited its highest level under this treatment. The NACA of *Bn* and *Ov* leaves increased to its highest level at 2 mM NH_4_^+^. The NACB of *Bn* and *Ov* leaves, respectively, reached their highest levels at 10 mM NH_4_ and 2 mM NH_4_^+^.

After adding 10 mM HCO_3_^−^, the *f*_A_ of *Bn* leaves was significantly increased only at 10 mM NH_4_, while the NACA significantly decreased across all ammonium treatment regimes. In contrast, the *f*_B_ and NACB significantly increased across all ammonium treatments. The *f*_A_ and NACA of *Ov* leaves significantly decreased, except for the treatment of 0 mM and 10 mM NH_4_^+^, while the *f*_B_ and NACB significantly increased between 0.5 mM and 2 mM NH_4_^+^.

### 3.9. The ATP Consumption of Leaves at Different Bicarbonate and Ammonium Treatments

In Table 5, at different bicarbonate and ammonium treatments, we qualified the ATP consumption of NO_3_^−^, NH_4_^+^, total inorganic nitrogen accumulation, and the difference in ATP consumption of total nitrogen assimilation after bicarbonate treatment in *Bn* and *Ov* leaves, labeled as ATPA, ATPB, ATP (A + B), and ΔATP (A + B), respectively. In B0 of *Bn*, the ATPA and ATP (A + B) decreased to the lowest level at 10 mM NH_4_^+^, where the ATPB increased to the highest level. As for *Ov* in B0, the ATPA increased to the highest level at the highest (20 mM) ammonium supply, where the ATPB decreased to the lowest level. However, the ATP (A + B) of *Ov* in B0 reached to the highest level at 2 mM NH_4_^+^.

After adding 10 mM HCO_3_^−^, the ATPA of *Bn* leaves in B0 was significantly increased only at 10 mM NH_4_^+^, while the ATPB significantly decreased except 20 mM NH_4_^+^, and the ATP (A + B) significantly decreased across all ammonium treatments. As for *Ov* of B10, the ATPA significantly decreased at other treatments except for 10 mM NH_4_^+^, but where the ATPB significantly increased. The ATP (A + B) of *Ov* in B10 only decreased significantly at a high ammonium level from 10 mM to 20 mM NH_4_^+^. ΔATP (A + B) of *Bn* and *Ov* decreased to the lowest level at the highest ammonium level (20 mM NH_4_^+^).

## 4. Discussion

### 4.1. Glucose Metabolism (EMP and PPP) and Growth in Bn and Ov Plants at Different Bicarbonate and Ammonium Levels

As it is known, high bicarbonate is not conducive to plant growth, leading to a decrease in the level of inorganic carbon and nitrogen metabolism of plants, which is manifested as a decrease in plant biomass [13,26]. In the B10 treatment, the growth of *Bn* and *Ov* plants reached its lowest values at 20 mM NH_4_^+^ (Table 2), which resulted in severe damage to plant cell membranes and a critical decline in the photosynthetic capacity and carbon assimilation (Figure 1 and Figure 2a). These responses can be attributed to osmotic stress caused by high bicarbonate and ammonium levels [27,28]. In this study, we found that bicarbonate significantly reduced the inhibition of *Ov* growth due to high ammonium. At 10 mM HCO_3_^−^, the biomass, photosynthesis, and leaf inorganic carbon content decreased in both *Bn* and *Ov* plants. However, the reduction in *Ov* was less severe compared to *Bn* (Table 5, and Figure 1a and Figure 2a), and the biomass of *Bn* continued to decline with the increase in ammonium supply. Still, the *Ov* biomass increased at 2 mM NH_4_^+^, indicating different responses of carbon assimilation capacity to bicarbonate and assimilation among various plant species. It has been shown that stomatal movement is positively correlated with plant photosynthesis [29,30], which supports our findings that high bicarbonate and ammonium inhibited Gs, thereby reducing photosynthesis in *Bn* and *Ov* plants. Moreover, the activities of carbon metabolism enzymes, such as Rubisco, SS, and PFK, decreased (Figure 2 and Figure 3) as a result of the reduced photosynthesis, which decreased carboxylation efficiency in plant cells [31,32]. However, compared to *Bn*, we observed that bicarbonate reduced the inhibitory effects of high ammonium on *Ov* plants’ growth. Bicarbonate significantly alleviated the Gs of *Ov* plants (Figure 1b). At the same time, the *Ov* plants exhibited an increased carbonic anhydrase (CA) activity, catalyzing the conversions of HCO_3_^−^ to CO_2_ and H_2_O to relieve stress in karst habitats [23,24]. In addition, the Tr and WUE of *Ov* plants were decreased to a lower extent compared to *Bn* (Figure 1d,e), which resulted in a significantly lower reduction in the Pn of *Ov* plants. Moreover, it was also attributed to a greater PPP and RC_RUBP_ of *Ov* plants (Figure 4d and Figure 5b). Under abiotic stress, it has been reported that the organic carbon of plant species results in a shift from EMP to PPP, in a percentage even higher than 50% [33,34]. PPP enhances NADPH production [35,36], promoting the cells’ metabolic activity, including the carbon- and nitrogen-metabolizing enzymes (Figure 1b). The shift to the PPP was also conducive to nitrate reduction and the formation of RuBP, which promoted the regenerative capacity of RuBP in the Calvin–Benson cycle. In *Ov* plants, the increase in PPP promoted excess light energy release in PSI and reduced the damage to the PSI reaction center proteins, which maintained the Pn and GC [12,37].

### 4.2. Differential Responses of Inorganic NO_3_^−^ and NH_4_^+^ to Bicarbonate and Ammonium in Bn and Ov Plants

It has been shown that the nitrogen/ammonium ratio is an essential factor affecting plant growth, photosynthesis, and yield [38]. In karst soils and under drought, high bicarbonate, and high pH habitats, bicarbonate regulates the nitrate/ammonium utilization and plants’ growth [5]. In addition, characterized as a “high nitrate and low ammonium” soil, the ammonium supply is beneficial to promote a virtuous nitrogen environment in karst soils [13]. Consequently, we evaluated the differential growth responses to bicarbonate and various ammonium levels of two *Brassicaceae* species. It has been reported that a high NR activity determines a higher inorganic N accumulation capacity. In this experiment, with the increase in NH_4_^+^ supply, we found that the inorganic N content of *Bn* and *Ov* leaves reached a maximum level at 2 mM NH_4_^+^, attributed to high NR activity and NAC [39]. Moreover, bicarbonate was more beneficial to the inorganic nitrogen assimilation of *Ov* plants, increasing their nitrate-reducing capacity and the capacity for carbon skeleton formation [14]. Otherwise, there was a different response of *Bn* and *Ov* leaves’ inorganic N content to bicarbonate and various ammonium levels. In this experiment, bicarbonate promoted, to a more significant extent, the leaf inorganic N content in *Ov* at 0.5 mM and 2 mM NH_4_^+^ compared to *Bn*. This might be attributed to the photosynthetic efficiency and growth of *Ov* [40,41]. Bicarbonate and ammonium inhibited the growth of *Bn*, leading to a decrease in leaf inorganic N accumulation. In contrast, at a low ammonium supply level, bicarbonate increased Gs in *Ov*, which promoted its Pn and growth (Table 1 and Figure 1). However, at the highest ammonium levels, the inorganic N content of *Bn* and *Ov* leaves decreased to its lowest level, attributed to growth inhibition (Table 1) [42], leading to the decline in inorganic N accumulation in both *Bn* and *Ov*.

Stable nitrogen isotope analysis can reveal the relationship between external nitrogen supply and plant nitrogen demand [43]. In this experiment, we calculated the “post-application” *δ*^15^N value of *Bn* and *Ov* leaf at bicarbonate and various ammonium level. High nitrate utilization determined the leaf *δ*^15^N value [44]. At low NH_4_^+^ levels (0.5 mM–2 mM NH_4_^+^), the *Bn* and *Ov* leaves mainly utilized NO_3_^−^, resulting in no significant change in the post-application *δ*^15^N value of the leaves. Above 0.5 mM NH_4_^+^, the ammonium utilization of *Bn* and *Ov* increased, decreasing the post-application *δ*^15^N value. Still, the *δ*^15^N value of *Bn* and *Ov* leaves showed an exceptional increase at the highest ammonium level (20 mM NH_4_^+^), which could be attributed to the inhibition of plant growth caused by excess ammonium. At the same time, the Δ^15^N value and *δ*^15^N value showed an opposite trend [45]. At 20 mM NH_4_^+^, the Δ^15^N value of *Bn* and *Ov* leaves reached its highest level, indicating that the excessive NH_4_^+^ might produce “NH_4_^+^ toxicity” due to plant growth inhibition. A high nitrogen assimilation capacity was reported to correspond to a high *δ*^15^N value [15]. In B10, the post-application *δ*^15^N value of *Bn* and *Ov* leaves increased significantly at 10 mM NH_4_^+^, indicating that bicarbonate and ammonium promoted inorganic N assimilation in *Bn* and *Ov* leaves. In contrast, the post-application *δ*^15^N value of *Bn* and *Ov* leaves decreased at 20 mM NH_4_^+^, probably attributed to the inhibition of nitrogen assimilation (Figure 3) and growth (Table 1) in *Bn* and *Ov*.

We further quantified the NO_3_^−^ and NH_4_^+^ utilization (*f*_A_ and *f*_B_) of *Bn* and *Ov* leaves based on the bidirectional nitrogen isotope tracing method, with the sum of *f*_A_ and *f*_B_ in the plant always at 100%. As there is no ammonium salt in the Hogland solution, we assumed that nitrate is the only nitrogen source for total inorganic nitrogen assimilation, so that the *f*_A_ of *Bn* and *Ov* is 100%, while *f*_B_ is 0 (0%). With an increase in NH_4_^+^, the *f*_A_ and NACA of *Bn* and *Ov* increased, but then it decreased at higher NH_4_^+^ concentrations. Still, the *f*_B_ and NACB showed the opposite trend, indicating that plants might preferentially utilize NO_3_^−^ at a low NH_4_^+^ level and NH_4_^+^ at a high NH_4_^+^ level. In B10, the *f*_B_ of *Bn* and *Ov* significantly increased, indicating that bicarbonate promoted NH_4_^+^ utilization in *Bn* and *Ov* plants. At the highest NH_4_^+^ (20 mM NH_4_^+^), the NACA and NACB significantly decreased in *Bn*, but the NACB increased in *Ov*. It has been shown that plant ammonium assimilation requires ATP and carbon sources [17]. Compared to *Bn*, bicarbonate promoted photosynthesis (Figure 1), the leaf carbon content (Figure 2), and the total glucose metabolism (Figure 4) of *Ov* plants, resulting in the promotion of NH_4_^+^ assimilation.

### 4.3. NH_4_^+^/NO_3_^−^ Utilization Determines Plant Growth Rather Than the Absolute Ammonium Levels or the Variable Sources of Total Nitrogen

Generally, NO_3_^−^ reduction and NH_4_^+^ assimilation, which dominate inorganic nitrogen assimilation in plant species, consume ATP [17]. It was reported that the assimilation of 1 mol NO_3_^−^ consumes 20 mol ATP, while the assimilation of 1 mol NH_4_^+^ consumes 5 mol ATP [21]. In this study, we calculated the NO_3_^−^ reduction, NH_4_^+^ assimilation, and the ATP consumption of total nitrogen assimilation, ATPA, ATPB, and ATP (A + B). As shown in Table 5, we found that ATPA was significantly higher than ATPB in both *Bn* and *Ov*, indicating that NO_3_^−^ reduction consumes more ATP than NH_4_^+^ assimilation, which is consistent with their metabolic demands. It has been demonstrated that optimal growth conditions determine plants’ inorganic nitrogen assimilation capacity [17,46]. In this experiment, B0, *Bn*, and *Ov* had the highest ATP at 2 mM NH_4_^+^, indicating a greatest inorganic N assimilation capacity, attributed to the optimal biomass production (Table 1). On the contrary, the ATP (A + B) of *Bn* and *Ov* reached their lowest levels at 20 mM NH_4_^+^, attributed to their significantly reduced growth. Compared to *Ov*, the ATPA, ATPB, and ATP (A + B) were greater in *Bn*, indicating a higher ATP consumption for total inorganic N assimilation, which resulted in a higher photosynthetic efficiency (Figure 1) and biomass (Table 1) of *Bn*. However, compared to *Bn*, we found that bicarbonate more significantly increased NH_4_^+^ assimilation in *Ov*, as evidenced by the higher ATPB values (part (b) in Table 5). In contrast, bicarbonate significantly reduced ATP (A + B) in *Bn*, except for the highest NH_4_^+^ level. At the same time, the ATP (A + B) showed no significant changes in *Ov*, which was attributed to bicarbonate more significantly inhibiting the photosynthesis and growth of *Bn*, resulting in an inorganic N assimilation reduction. In addition, we found that bicarbonate more significantly reduced the ATP consumption for inorganic nitrogen assimilation in *Ov* than that in *Bn*. Apparently, in B10, the ΔATP (A + B) of *Bn* was higher compared to *Ov*, which indicated that bicarbonate resulted in increased ATP consumption in *Bn* for nitrogen assimilation compared to *Ov*, especially at low NH_4_^+^ levels (0.5 mM–2 mM NH_4_^+^). For example, at 0.5 mM NH_4_^+^, the ΔATP (A + B) of *Ov* was −4.50 mM, indicating that 10 mM bicarbonate reduced the ATP levels for total nitrogen assimilation by 2.85 mM, perhaps implying that bicarbonate reduced the ATP required for total inorganic nitrogen assimilation in *Ov* by 18.38%. Increasing ATP consumption resulted in the inhibition of plant growth [1]. In B10, the reduction in *Ov*’s growth was significantly lower compared to that of *Bn* (Table 1), indicating that bicarbonate more significantly promoted the growth of *Ov*. This result further validates that karst-adaptable plant species, such as *Ov*, exhibit a “predatory uptake of NH_4_^+^” to maintain growth under the conditions of drought, high bicarbonate (high pH), abundant NO_3_^−^, and bare NH_4_^+^ soil habitats [13].

### 4.4. Correlation of Glucose Metabolism and Nitrogen Utilization in Bn and Ov Plants under Bicarbonate and Variable Ammonium

An excessive ammonium supply may lead to acidic stress, which results in the “ammonium toxicity” observed in plants [16]. In this study, after treatment with over 2 mM NH_4_^+^, the increased ammonium supply reduced the growth of *Bn* and *Ov* plants (Table 1 and Figure 1), decreasing the inorganic carbon and nitrogen accumulation capacity. This was attributed to the high ammonium content, which has been shown to lead to large amounts of H^+^ consumption and the production of OH^−^, inhibiting photosynthesis and inorganic nitrogen assimilation in plants [47]. Therefore, an ineffective ammonium cycle is detrimental to plant growth. In this study, bicarbonate significantly alleviated the high ammonium inhibition of growth in *Ov* plants compared to *Bn* plants, which was attributed to the higher glucose metabolic capacity (Figure 4 and Figure 5). In addition, a higher correlation was observed in *Ov* among EMP, NACA and NACB, PPP, and *f*_B_ compared to those in *Bn* (Figure 6). It is well known that inorganic nitrogen assimilation requires energy and carbon skeletons derived from photosynthesis and glucose metabolism [5,12]. Under bicarbonate and high ammonium conditions, *Ov* maintained a higher PPP (Figure 4d) and ammonium utilization than *Bn* (part (b) in Table 4). Consequently, this resulted in a reduction in the ineffective ammonium cycle and the promotion of inorganic nitrogen assimilation and photosynthesis. Furthermore, it reflected the synergistic positive interaction of inorganic carbon and nitrogen assimilation in *Ov* plants under bicarbonate and high ammonium stress. These results suggest that, in karst habitats, photosynthetic oxygen release not only indicates water photolysis but is also involved in both water photolysis and bicarbonate photolysis (Figure 6c,d), providing electrons/H^+^ for ADP/NADP^+^ to produce ATP/NADPH for glucose metabolism, including the EMP, PPP, and TCA cycles in plants. Plants meet their nitrogen requirements mainly through nitrate reduction, which consumes energy. Therefore, the presence of low-level ammonium salt can reduce energy loss due to plant nitrogen assimilation and promote plant growth. In addition, low levels of ammonium salt can alleviate the high pH environment produced by high bicarbonate and maintain the homeostasis of the physiological functions of the roots. However, high ammonium is not conducive to plant growth, increasing plant energy expenditure. Fortunately, in karst habitats, bicarbonate is an effective carbon source, and plants may supplement water and carbon sources through “bicarbonate photolysis” to reduce energy loss and maintain plant growth. “Bicarbonate photolysis” appears to be more pronounced in karst-adaptable plants due to their efficient bicarbonate use capacity (BUC) [10], which results in *Ov* (a karst-adaptable plant species) having higher efficiency in bicarbonate photolysis, promoting ATP and NAPDH production to maintain its carbon and nitrogen metabolism cycle, leading to the enhancement of EMP, PPP, NACA, and NACB under high ammonium stress (Figure 6d). On the other hand, *Bn* plants have a lower BUC, resulting in less ATP and NADPH produced by bicarbonate photolysis (Figure 6c) and a larger decrease in inorganic carbon and nitrogen utilization efficiency under bicarbonate and high ammonium stress conditions.

## 5. Conclusions

Based on the bidirectional nitrogen isotope tracing method results and the quantification of EMP and PPP contribution, bicarbonate and ammonium inhibited the inorganic carbon and nitrogen utilization in *Bn* plants grown in simulated karst habitats. Compared with *Bn*, bicarbonate promoted inorganic carbon and nitrogen utilization in *Ov* plants at low ammonium levels, leading to an increase in photosynthesis, carbon- and nitrogen-metabolizing enzyme activities, PPP, nitrate/ammonium utilization, and total inorganic nitrogen assimilation capacity. Moreover, bicarbonate significantly reduced the growth inhibition of *Ov* plants by high ammonium, resulting in improved PPP, RC_RUBP_, and ammonium utilization to maintain growth. Our results indicate that bicarbonate leads to a shift from the EMP to the PPP, improving the ammonium utilization in *Ov* under high ammonium stress in karst habitats.

## Figures and Tables

**Figure 1 plants-12-03095-f001:**
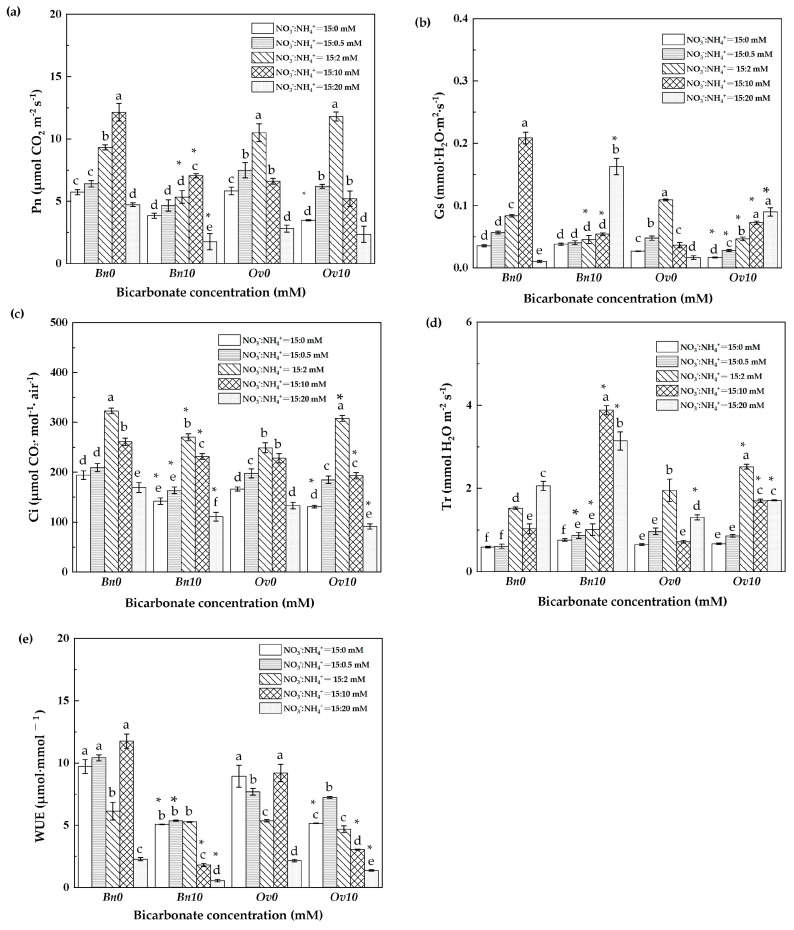
Different responses of photosynthesis to bicarbonate and ammonium in *Bn* and *Ov*. *Bn*: *Brassica napus*; *Ov*: *Orychophragmus violaceus*. B0: 0 mM HCO_3_^−^; B10: 10 mM HCO_3_^−^; NO_3_^−^ concentration: 15 mM; NH_4_^+^: 0/0.5/2/10/20 mM. (**a**) Pn: net photosynthetic rate, μmol CO_2_·m^2^·s^−1^; (**b**) Gs: stomatal conductance, mmol H_2_O·m^2^·s^−1^; (**c**) Ci: intercellular carbon dioxide concentration, μmol CO_2_·mol·air ^−1^; (**d**) Tr: transpiration, μmol H_2_O·m^2^·s^−1^; (**e**) WUE: water use efficiency, μmol·mmol^−1^. “*” represents a *p*-value less than 5% level of significance. Each value represents the mean ± SD (*n* = 3), and diverse letters in each value are significantly different by ANOVA (*p* ≤ 0.05).

**Figure 2 plants-12-03095-f002:**
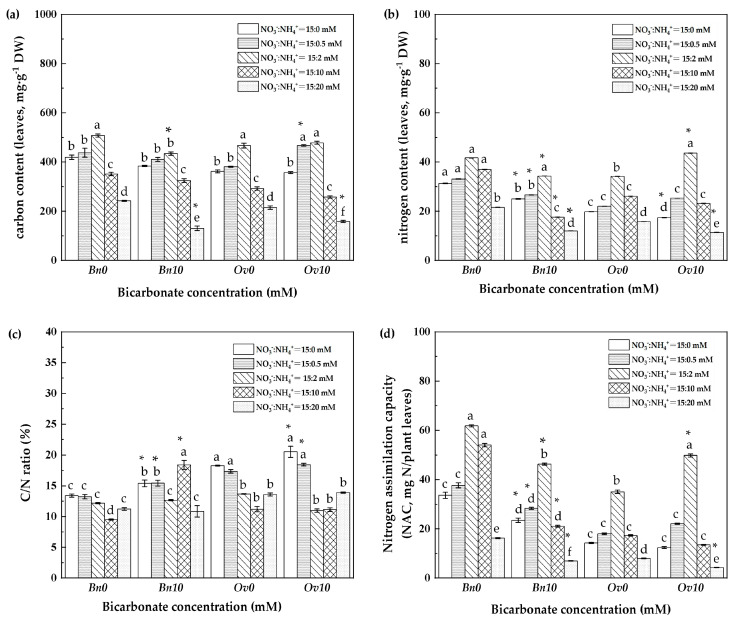
The carbon/nitrogen component in leaves between *Bn* and *Ov* at different bicarbonate and ammonium levels. *Bn*: *Brassica napus*; *Ov*: *Orychophragmus violaceus*. B0: 0 mM HCO_3_^−^; B10: 10 mM HCO_3_^−^; NO_3_^−^ concentration: 15 mM; NH_4_^+^: 0/0.5/2/10/20 mM. (**a**) Carbon content of leaves, mg·g^−1^ DW; (**b**) Nitrogen content of leaves, s, mg·g^−1^ DW; (**c**) C/N ratio, %; (**d**) NAC: nitrogen assimilation capacity, %. “*” represents a *p*-value less than 5% level of significance. Each value represents the mean ± SD (*n* = 3), and diverse letters in each value are significantly different by ANOVA (*p* ≤ 0.05).

**Figure 3 plants-12-03095-f003:**
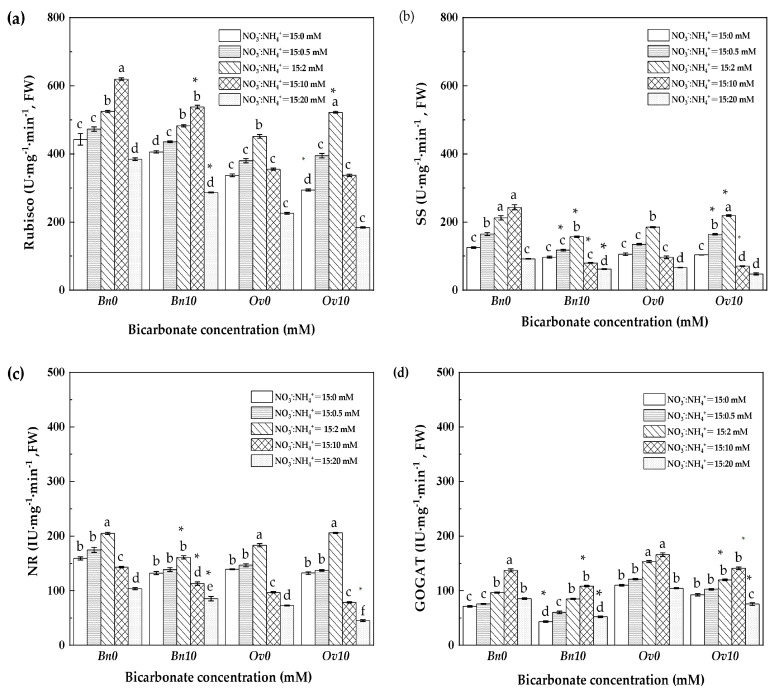
Rubisco, SS, NR, and GOGAT activities of *Bn* and *Ov* under different bicarbonate and ammonium supplies. *Bn*: *Brassica napus*; *Ov*: *Orychophragmus violaceus*. B0: 0 mM HCO_3_^−^; B10: 10 mM HCO_3_^−^; NO_3_^−^ concentration: 15 mM; NH_4_^+^: 0/0.5/2/10/20 mM. (**a**) Rubisco: ribulose bisphosphate carboxylase oxygenase, U·mg^−1^·min^−1^; (**b**) SS: sucrose synthetase, U·mg^−1^·min^−1^; (**c**) NR: nitrate reductase, IU·mg^−1^·min^−1^; (**d**) GOGAT: glutamate synthase, IU·mg^−1^·min^−1^. “*” represents a *p*-value less than 5% level of significance. Each value represents the mean ± SD (*n* = 3), and diverse letters in each value are significantly different by ANOVA (*p* < 0.05).

**Figure 4 plants-12-03095-f004:**
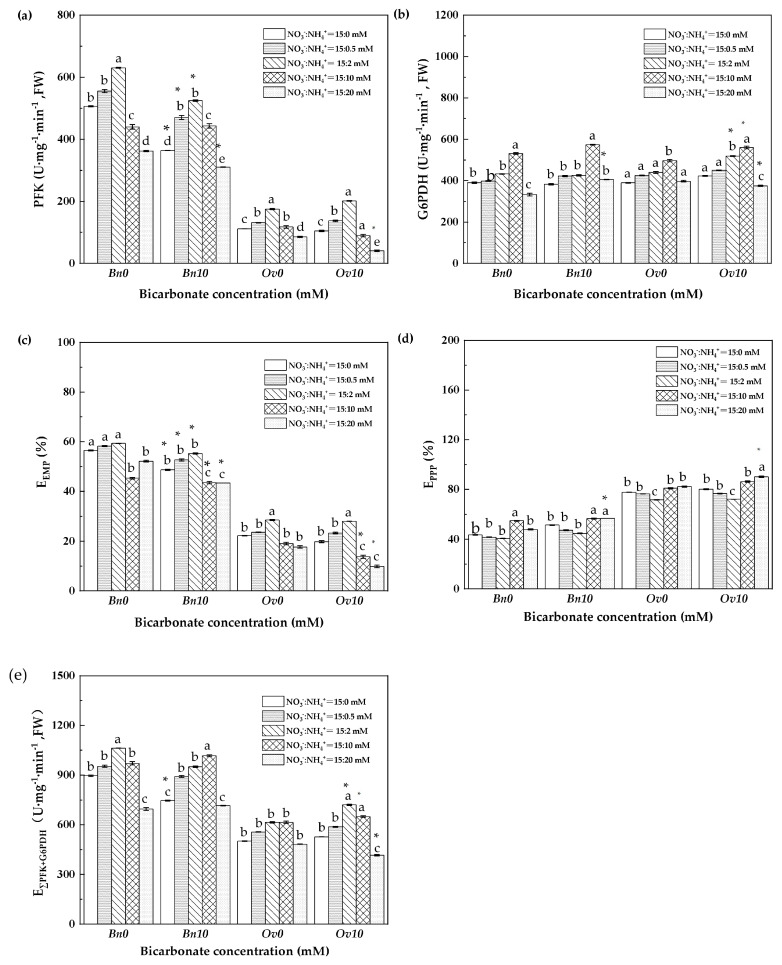
The glycolysis pathway and the pentose phosphate pathway of *Bn* and *Ov* at different bicarbonate and ammonium levels. *Bn*: *Brassica napus*; *Ov*: *Orychophragmus violaceus*. B0: 0 mM HCO_3_^−^; B10: 10 mM HCO_3_; NO_3_^−^ concentration: 15 mM; NH_4_^+^ concentration: 0/0.5/2/10/20 mM. (**a**) PFK: Phosphofructokinase, U·mg^−1^·min^−1^; (**b**) G6PDH: Glucose-6-phosphate dehydrogenase, U·mg^−1^·min^−1^; (**c**) E_EMP_: the contribution of the glycolytic pathway, %; (**d**) E_PPP_: the contribution of the pentose phosphate pathway, %; (**e**) E∑_EMP+PPP_: total glucose metabolism, U·mg^−1^·min^−1^. “*” represents a *p*-value less than 5% level of significance. Each value represents the mean ± SD (*n* = 3), and diverse letters in each value are significantly different by ANOVA (*p* < 0.05).

**Figure 5 plants-12-03095-f005:**
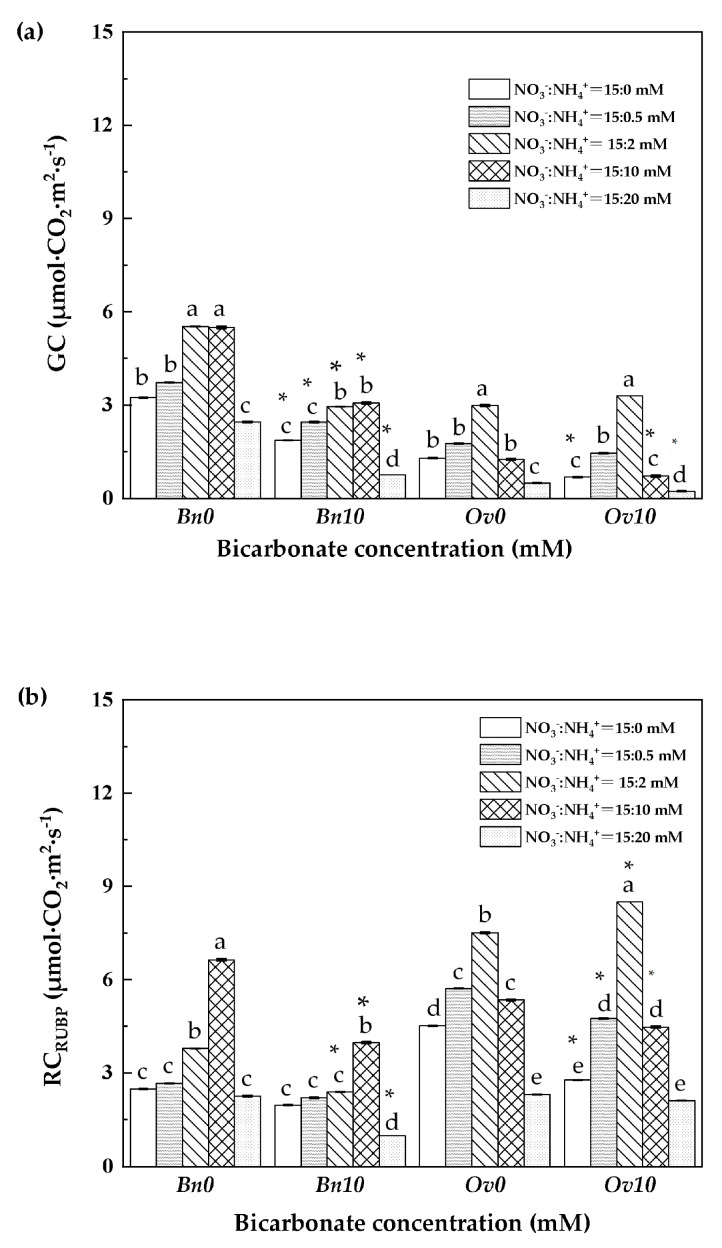
Growth capacity and regeneration capacity of RUBP in *Bn* and *Ov* at different bicarbonate and ammonium levels. *Bn*: *Brassica napus; Ov*: *Orychophragmus violaceus*. B0: 0 mM HCO_3_^−^; B10: 10 mM HCO_3_; NO_3_^−^ concentration: 15 mM; NH_4_^+^ concentration: 0/0.5/2/10/20 mM. (**a**) GC:Growth capacity, μmol CO_2_·m^2^·s^−1^; (**b**) RC_RUBP_: the regeneration capacity of Ribulose-1,5-bisphosphate, μmol CO_2_·m^2^·s^−1^. “*” represents a *p*-value less than 5% level of significance. Each value represents the mean ± SD (*n* = 3), and diverse letters in each value are significantly different by ANOVA (*p* ≤ 0.05).

**Figure 6 plants-12-03095-f006:**
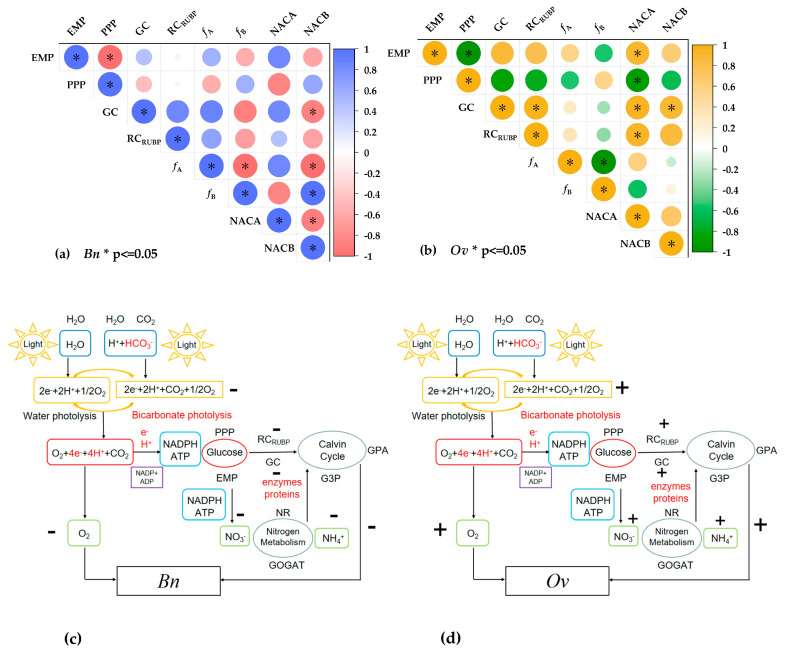
The correlation between glucose metabolism and inorganic nitrogen use of *Bn* and *Ov* at different bicarbonate and ammonium levels. *Bn*: *Brassia naps*; *Ov*: *OTychoplhragmus violaceu*. B0: 0 mM HCO_3_^−^; B10:10 mM HCO_3_^−^; NO_3_^−^; concentration: 15 mM; NH_4_^+^ concentration: 0/0.5/2/10/20 mM. (**a**) The correlation between glucose metabolism and inorganic nitrogen use of *Bn*; (**b**) the correlation between glucose metabolism and inorganic nitrogen use of *Ov*; (**c**) the inorganic carbon and nitrogen mechanisms in *Bn* at different bicarbonate and ammonium levels; (**d**) the inorganic carbon and nitrogen mechanism in *Ov* at different bicarbonate and ammonium levels. “*” represents a *p*-value less than 5% level of significance.

**Table 1 plants-12-03095-t001:** Different bicarbonate and ammonium treatments.

Group	Reagent	Content
B0	bicarbonate	0 mM
B10	bicarbonate	10 mM
NO^3−^:NH^4+^	δ^15^N_H_:22.72‰; δ^15^N_L_:12.7‰	15:0 mM
NO^3−^:NH^4+^	δ^15^N_H_:22.72‰; δ^15^N_L_:12.7‰	15:0.5 mM
NO^3−^:NH^4+^	δ^15^N_H_:22.72‰; δ^15^N_L_:12.7‰	15:2 mM
NO^3−^:NH^4+^	δ^15^N_H_:22.72‰; δ^15^N_L_:12.7‰	15:10 mM
NO^3−^:NH^4+^	δ^15^N_H_:22.72‰; δ^15^N_L_:12.7‰	15:20 mM
pH	KOH	8.30 ± 0.05
Simulated drought	PEG6000	10 g·L^−1^, −0.21 Mpa

**Table 2 plants-12-03095-t002:** Growth of *Bn* and *Ov* under different ammonium and bicarbonate levels.

(a)						
Group	Plant	*Bn* (mg·Plant^−1^, DW)				
	NO_3_^−^:NH_4_^+^	15 mM:0 mM	15 mM:0.5 mM	15 mM:2 mM	15 mM:10 mM	15 mM:20 mM
Root Dw:mg	B0	212.01 ± 1.49 d	223.13 ± 1.13 d	296.63 ± 4.86 b	322.53 ± 4.17 a	105.35 ± 1.75 d
	B10	180.34 ± 5.12 d	196.89 ± 4.9 d	205.97 ± 5.18 c	266.83 ± 4.48 c	73.23 ± 1.14 e
Stem Dw:mg	B0	335.36 ± 4.28 c	352.56 ± 1.95 c	418.54 ± 4.65 b	465.58 ± 7.68 a	154.59 ± 2.53 d
	B10	301.07 ± 1.13 d	299.23 ± 1.45 d	324.73 ± 4.77 c	477.68 ± 3.1 a	101.1 ± 1.88 f
Leaves DW:mg	B0	528.4 ± 2.43 e	562.46 ± 2.3 d	692.95 ± 4.73 b	744.15 ± 4.54 a	493.24 ± 2.1 f
	B10	460.48 ± 3.22 f	534.45 ± 4.28 e	605.94 ± 3.83 c	700.63 ± 5.43 b	406.82 ± 2.36 g
Biomass Dw:mg	B0	1075.76 ± 6.72 c	1139.15 ± 3.17 c	1459.32 ± 11.47 a	1481.05 ± 13.88 a	753.19 ± 5.8 e
	B10	941.89 ± 6.3 d	1030.57 ± 11.44 c	1065.16 ± 12.16 c	1350.45 ± 6.37 b	581.15 ± 4.25 f
**(b)**						
**Group**	**Plant**	***Ov*** **(mg·Plant^−1^, DW)**				
	**NO_3_^−^:NH_4_^+^**	**15 mM:0 mM**	**15 mM:0.5 mM**	**15 mM:2 mM**	**15 mM:10 mM**	**15 mM:20 mM**
Root Dw:mg	B0	91.67 ± 0.95 c	112.35 ± 2.45 b	159.14 ± 2.59 a	74.85 ± 2.09 d	51.42 ± 0.57 e
	B10	101.69 ± 1.36 c	124.35 ± 2.25 b	179.97 ± 3.62 a	59.63 ± 0.58 e	32.33 ± 0.8 f
Stem Dw:mg	B0	160.44 ± 9.81 d	202.31 ± 2.6 c	278.31 ± 4.48 b	138.45 ± 3.02 e	92.88 ± 1.53 g
	B10	162.28 ± 2.02 d	208.94 ± 2.02 c	317.99 ± 0.83 a	113.13 ± 2.44 f	62.87 ± 2.79 h
Leaves DW:mg	B0	468.69 ± 4.73 d	502.92 ± 2.43 bc	588.41 ± 6.84 b	451.29 ± 9.8 d	360.02 ± 3.28 f
	B10	447.47 ± 5.65 d	538.68 ± 1.42 b	643.08 ± 6.63 a	409.65 ± 2.55 e	280.66 ± 1.58 g
Biomass Dw:mg	B0	720.8 ± 8.09 d	818.58 ± 7.85 c	1025.85 ± 9.15 a	664.59 ± 14.82 e	504.32 ± 4.59 g
	B10	711.45 ± 7 d	871.96 ± 1.23 b	1141.05 ± 10.98 a	582.4 ± 5.3 f	375.86 ± 2.67 h

*Bn*: *Brassica napus*; *Ov*: *Orychophragmus violaceus*. B0: 0 mM HCO_3_^−^; B10: 10 mM HCO_3_^−^; NO_3_^−^ concentration: 15 mM; NH_4_^+^: 0/0.5/2/10/20 mM. Each value represents the mean ± SD (*n* = 3), and diverse letters in each value are significantly different by ANOVA (*p* ≤ 0.05).

**Table 3 plants-12-03095-t003:** The *δ*^15^N and Δ^15^N values of *Bn* and *Ov* leaves under different bicarbonate and ammonium levels. Growth of *Bn* and *Ov* under different ammonium and bicarbonate levels.

(a)						
	NO_3_^−^:NH_4_^+^	15 mM:0 mM	15 mM:0.5 mM	15 mM:2 mM	15 mM:10 mM	15 mM:20 mM
*δ*^15^N_NEW_ (H, ‰)	*Bn*0	20.37 ± 0.66 b	21.05 ± 0.4 b	19.12 ± 0.13 b	13.92 ± 0.11 c	28.64 ± 0.88 a
	*Bn*10	29.16 ± 1.76 a	23.34 ± 0.4 b	18.96 ± 0.15 b	24.36 ± 1.32 b	−48.8 ± 3.72 d
	*Ov*0	22.91 ± 0.58 c	20.51 ± 0.22 c	16.34 ± 0.07 d	16.03 ± 0.1 d	49.08 ± 2.84 a
	*Ov*10	30.25 ± 1.31 b	17.98 ± 0.2 c	15.72 ± 0.55 d	19.97 ± 0.42 c	−31.8 ± 2.32 e
**(b)**						
	**NO_3_^−^:NH_4_^+^**	**15 mM:0 mM**	**15 mM:0.5 mM**	**15 mM:2 mM**	**15 mM:10 mM**	**15 mM:20 mM**
*δ*^15^N_NEW_ (L, ‰)	*Bn*0	10.41 ± 0.6 b	9.22 ± 0.13 b	7.11 ± 0.06 c	5.71 ± 0.21 c	18.05 ± 0.6 a
	*Bn*10	18.81 ± 1.44 a	12.23 ± 0.41 b	8.27 ± 0.16 c	13.77 ± 0.54 b	−38.66 ± 2.72 d
	*Ov*0	12.68 ± 0.3 c	10.75 ± 0.46 c	7.41 ± 0.15 d	7.28 ± 0.12 d	29.16 ± 1.58 a
	*Ov*10	16.43 ± 0.65 b	8.98 ± 0.23 d	7.13 ± 0.08 d	8.42 ± 0.33 d	−25.44 ± 1.73 e
**(c)**						
	**NO_3_^−^:NH_4_^+^**	**15 mM:0 mM**	**15 mM:0.5 mM**	**15 mM:2 mM**	**15 mM:10 mM**	**15 mM:20 mM**
Δ^15^N_NEW_ (H, ‰)	*Bn*0	−12.29 ± 0.66 c	−12.97 ± 0.4 c	−11.04 ± 0.13 c	−5.84 ± 0.11 b	−20.56 ± 0.88 e
	*Bn*10	−21.08 ± 1.76 e	−15.26 ± 0.4 d	−10.88 ± 0.15 c	−16.28 ± 1.32 c	56.88 ± 3.72 a
	*Ov*0	−14.83 ± 0.58 d	−12.43 ± 0.22 c	−8.26 ± 0.07 b	−7.95 ± 0.1 b	−41 ± 2.84 f
	*Ov*10	−22.17 ± 1.31 e	−9.9 ± 0.2 c	−7.64 ± 0.55 b	−11.89 ± 0.42 d	39.88 ± 2.32 a
**(d)**						
	**NO_3_^−^:NH_4_^+^**	**15 mM:0 mM**	**15 mM:0.5 mM**	**15 mM:2 mM**	**15 mM:10 mM**	**15 mM:20 mM**
Δ^15^N_NEW_ (L, ‰)	*Bn*0	−2.33 ± 0.6 f	−1.14 ± 0.13 e	0.97 ± 0.06 c	2.37 ± 0.21 b	−9.97 ± 0.6 h
	*Bn*10	−10.73 ± 1.44 h	−4.15 ± 0.41 g	−0.19 ± 0.16 d	−5.69 ± 0.54 g	46.74 ± 2.72 a
	*Ov*0	−4.6 ± 0.3 e	−2.67 ± 0.46 b	0.67 ± 0.15 b	0.8 ± 0.12 b	−21.08 ± 1.58 g
	*Ov*10	−8.35 ± 0.65 f	−0.9 ± 0.23 c	0.95 ± 0.08 b	−0.34 ± 0.33 c	33.52 ± 1.73 a

*Bn*: *Brassica napus*; *Ov*: *Orychophragmus violaceus*. B0: 0 mM HCO_3_^−^; B10: 10 mM HCO_3_; NO_3_^−^ concentration: 15 mM; NH_4_^+^ concentration: 0/0.5/2/10/20 mM. (a) The *δ*^15^ N value with a high label of *Bn* and *Ov* leaves, ‰; (b) the *δ*^15^ N value with a light label of *Bn* and *Ov* leaves, ‰; (c) the Δ^15^N value with a high label of *Bn* and *Ov* leaves, ‰; (d) the Δ^15^N value with a light label of *Bn* and *Ov* leaves, ‰. Each value represents the mean ± SD (*n* = 3), and diverse letters in each value are significantly different by ANOVA (*p* ≤ 0.05).

**Table 4 plants-12-03095-t004:** Growth of *Bn* and *Ov* under different ammonium and bicarbonate levels. The utilization of NO_3_^−^/NH_4_^+^, NACA, and NACB in *Bn* and *Ov* at different bicarbonate and ammonium levels.

(a)						
	NO_3_^−^:NH_4_^+^	15 mM:0 mM	15 mM:0.5 mM	15 mM:2 mM	15 mM:10 mM	15 mM:20 mM
*f*_A_ (NO_3_^−^, %)	*Bn*0	100 b	119 ± 2 a	121 ± 3 a	82 ± 2 c	106 ± 4 b
	*Bn*10	100 b	108 ± 5 b	103 ± 3 b	102 ± 3 b	−98 ± 11 d
	*Ov*0	100 b	95 ± 5 b	87 ± 3 b	86 ± 2 b	195 ± 16 a
	*Ov*10	100 b	65 ± 4 c	62 ± 6 c	84 ± 6 b	−46 ± 4 d
**(b)**						
	**NO_3_^−^:NH_4_^+^**	**15 mM:0 mM**	**15 mM:0.5 mM**	**15 mM:2 mM**	**15 mM:10 mM**	**15 mM:20 mM**
*f*_B_ (NH_4_^+^, %)	*Bn*0	0 c	−19 ± 2 e	−21 ± 0.3 e	18 ± 2 b	−6 ± 4 d
	*Bn*10	0 c	−8 ± 5 d	−3 ± 0.3 c	−2 ± 0.3 c	198 ± 11 a
	*Ov*0	0 e	5 ± 0.5 d	13 ± 0.3 c	14 ± 2 c	−95 ± 16 f
	*Ov*10	0 e	35 ± 4 b	38 ± 0.6 b	16 ± 6 c	146 ± 4 a
**(c)**						
	**NO_3_^−^:NH_4_^+^**	**15 mM:0 mM**	**15 mM:0.5 mM**	**15 mM:2 mM**	**15 mM:10 mM**	**15 mM:20 mM**
NACA (NO_3_^−^, %)	*Bn*0	33.64 ± 1.25 c	44.79 ± 0.82 b	74.48 ± 1.22 a	44.52 ± 0.83 b	17.3 ± 0.57 e
	*Bn*10	23.51 ± 0.91 d	30.5 ± 1.93 c	47.85 ± 0.99 b	21.54 ± 0.44 e	−6.86 ± 0.91 f
	*Ov*0	14.27 ± 0.26 b	17.18 ± 1.12 b	30.66 ± 1.41 a	14.84 ± 0.57 b	15.54 ± 1.19 b
	*Ov*10	12.4 ± 0.4 b	14.4 ± 0.79 b	31.01 ± 2.79 a	11.29 ± 0.72 b	−1.97 ± 0.22 c
**(d)**						
	**NO_3_^−^:NH_4_^+^**	**15 mM:0 mM**	**15 mM:0.5 mM**	**15 mM:2 mM**	**15 mM:10 mM**	**15 mM:20 mM**
NACB (NH_4_^+^, %)	*Bn*0	0 c	−7.09 ± 0.53 h	−12.71 ± 1.52 i	9.47 ± 1.07 b	−1.04 ± 0.67 e
	*Bn*10	0 c	−2.15 ± 1.52 g	−1.54 ± 1.32 f	−0.46 ± 0.7 d	13.84 ± 1.03 a
	*Ov*0	0 f	0.81 ± 0.09 e	4.41 ± 0.09 c	2.51 ± 0.32 d	−7.57 ± 1.24 g
	*Ov*10	0 f	7.69 ± 0.84 b	18.79 ± 3.32 a	2.19 ± 0.76 d	6.26 ± 0.31 b

*Bn*: *Brassica napus*; *Ov*: *Orychophragmus violaceus*. B0: 0 mM HCO_3_^−^; B10: 10 mM HCO_3_; NO_3_^−^ concentration: 15 mM; NH_4_^+^ concentration: 0/0.5/2/10/20 mM. (a) *f*_A_: the contribution of nitrate, %; (b) *f*_B_: the contribution of ammonium, %; (c) NACA: the contribution of nitrate to total nitrogen accumulation capacity, %; (d) NACB: the contribution of ammonium to total nitrogen accumulation capacity, %. Each value represents the mean ± SD (*n* = 3), and diverse letters in each value are significantly different by ANOVA (*p* < 0.05).

**Table 5 plants-12-03095-t005:** The ATP consumption of ammonium assimilation in *Bn* and *Ov* at bicarbonate and variable ammonium levels.

(a)						
	NO_3_^−^:NH_4_^+^	15 mM:0 mM	15 mM:0.5 mM	15 mM:2 mM	15 mM:10 mM	15 mM:20 mM
ATPA (NO_3_^−^, mM)	*Bn*0	0 c	−7.09 ± 0.53 h	−12.71 ± 1.52 i	9.47 ± 1.07 b	−1.04 ± 0.67 e
	*Bn*10	0 c	−2.15 ± 1.52 g	−1.54 ± 1.32 f	−0.46 ± 0.7 d	13.84 ± 1.03 a
	*Ov*0	0 f	0.81 ± 0.09 e	4.41 ± 0.09 c	2.51 ± 0.32 d	−7.57 ± 1.24 g
	*Ov*10	0 f	7.69 ± 0.84 b	18.79 ± 3.32 a	2.19 ± 0.76 d	6.26 ± 0.31 b
**(b)**						
	**NO_3_^−^:NH_4_^+^**	**15 mM:0 mM**	**15 mM:0.5 mM**	**15 mM:2 mM**	**15 mM:10 mM**	**15 mM:20 mM**
ATPB (NH_4_^+^, mM)	*Bn*0	0 c	−0.95 ± 0.03 d	−1.05 ± 0.08 d	0.9 ± 0.05 b	−0.3 ± 0.03 c
	*Bn*10	0 c	−0.4 ± 0.08 c	−0.15 ± 0.07 c	−0.1 ± 0.04 c	9.9 ± 0.05 a
	*Ov*0	0 d	0.25 ± 0 d	0.65 ± 0 c	0.7 ± 0.02 c	−4.75 ± 0.06 e
	*Ov*10	0 d	1.75 ± 0.04 b	1.9 ± 0.17 b	0.8 ± 0.04 c	7.3 ± 0.02 a
**(c)**						
	**NO_3_^−^:NH_4_^+^**	**15 mM:0 mM**	**15 mM:0.5 mM**	**15 mM:2 mM**	**15 mM:10 mM**	**15 mM:20 mM**
ATP (A + B, mM)	*Bn*0	33.64 ± 1.25 c	44.79 ± 0.82 b	74.48 ± 1.22 a	44.52 ± 0.83 b	17.3 ± 0.57 e
	*Bn*10	23.51 ± 0.91 d	30.5 ± 1.93 c	47.85 ± 0.99 b	21.54 ± 0.44 d	−6.86 ± 0.91 f
	*Ov*0	14.27 ± 0.26 b	17.18 ± 1.12 b	30.66 ± 1.41 a	14.84 ± 0.57 b	15.54 ± 1.19 b
	*Ov*10	12.4 ± 0.4 c	14.4 ± 0.79 b	31.01 ± 2.79 a	11.29 ± 0.72 c	−1.97 ± 0.22 d
**(d)**						
	**NO_3_^−^:NH_4_^+^**	**15 mM:0 mM**	**15 mM:0.5 mM**	**15 mM:2 mM**	**15 mM:10 mM**	**15 mM:20 mM**
ΔATP (A + B, mM)	*Bn*	0 b	−1.65 c	−2.70 d	3.00 a	−30.60 f
	*Ov*	0 b	−4.50 e	−3.75 e	−0.30 b	−36.15 g

*Bn*: *Brassica napus*; *Ov*: *Orychophragmus violaceus*. B0: 0 mM HCO_3_^−^; B10: 10 mM HCO_3_; NO_3_^−^ concentration: 15 mM; NH_4_^+^ concentration: 0/0.5/2/10/20 mM. (a) ATPA: the ATP consumption of nitrate, mM; (b) ATPB: the ATP consumption of ammonium, mM; (c) ATP (A + B): the ATP consumption of total nitrogen assimilation, mM; (d) ΔATP (A + B) = ATP (A + B, B10) − ATP (A + B, B0), the difference in the ATP consumption of the total nitrogen assimilation, mM. Each value represents the mean ± SD (*n* = 3), and diverse letters in each value are significantly different by ANOVA (*p* ≤ 0.05).

## Data Availability

The data presented in this study are available up +on request from the corresponding author.

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
