# Peer review of "Quantification of Glucose Metabolism and Nitrogen Utilization in Two Brassicaceae Species under Bicarbonate and Variable Ammonium Soil Conditions"

_plants, 2023, doi:10.3390/plants12173095_

Round 1
Reviewer 1 Report
Moderate corrections are required.
1. Try to make italics in the Latin name in the title
2. Why did You choose the ANOVA test, not Bonferoni test?
3. In 2.11 no superscript and subscript in the structural formula please write according to standards
There are tenses mixed resulting in unclear sentences. The values stick to units, especially in the Introduction and methodology. Please correct
Author Response
Question 1: Try to make italics in the Latin name in the title?
Response: Thanks for your valuable comments. We have revised the title, see line 3.
Question 2: Why did You choose the ANOVA test, not Bonferoni test?
Response: Thanks for your kind comments. It has been reported that Bonferroni is a multiple testing method, which is calculated by dividing the a level of a multiple test into several single test a levels, so as to reduce the error rate of the test.The advantage of Bonferroni is that it can effectively detect the difference of multiple samples, the disadvantage is that it is prone to false positives, that is, to judge the real non-differences of the samples as having differences as well, and it is suitable for It is suitable for testing differences in means in multiple samples. Bonferroni correction is more conservative and prone to incorrectly accepting the null hypothesis. The advantages of the ANOVA method are: (1) it is not limited by the number of statistical groups, can accept a large sample of statistical quantities for multiple comparisons, can fully utilize the data provided by the test to estimate the experimental error, can separate the effects of the factors on the test indexes from the experimental error, it is a quantitative analysis method, comparable, high analytical precision; (2) ANOVA can examine the interaction of multiple factors. Hence, we chose ANOVA test to analyse data in our experiment.
Question 3: In 2.11 no superscript and subscript in the structural formula please write according to standards.
Response: Thanks for the reviewer's objective questions. We have revised the content, see line 313.
As mentioned above, we tried our best to improve the manuscript and made some changes in the revised manuscript. These changes will not influence the content and framework of the manuscript.
We appreciate for reviewers and editors' warm work earnestly, and hope that the correction will meet with approval.
Author Response
Comments: The submitted paper entitled 'Quantification of glucose metabolism and nitrogen utilisation in two Brassicaceae species under bicarbonate and variable ammonium soil conditions' proved to be a really interesting study, especially given the innovation of the type of species chosen for the study as well as the soil in which to test the aforementioned hypothesis.
Some points to review:
However, despite the amount of data collected and the appropriate materials and methods for the type of study, I feel compelled to ask for a thorough review of the results. I found the results in all the paragraphs to be very general, far too general, and I would ask that at least some of the 'significant' (letter-based) results be discussed in order to make the work clearer and not have readers waste their time going through the tables and/or figures to find what is being discussed and whether it is true or not.
Response: We would like to thank you for your careful reading, helpful comments, and constructive suggestions, which has significantly improved the presentation of our manuscript. Based on your valuable suggestion, we have carefully considered all comments from the reviewers and revised our manuscript accordingly. It needs to be understood by the reviewers that due to the many experiments set up (20 groups), we have developed a detailed discussion of the results. For ease of reading, firstly we list the maximum and minimum values so that they can be compared with other data. In addition, we have also listed the "significant" results in order to enhance the understanding of reviewers and readers, for example, section 3.6 (lines 434-439). Thus, most of the results are expressed in this way of our experiments. Definitely, we appreciate for your valuable suggestions, and sincerely wish you will agree with our presentation to the main points in this article. The manuscript has also been double-checked, and the typos and grammar errors we found have been corrected(certificate see as attachment). In the following section, we summarize our responses to each comment from the reviewers. We believe that our responses have well addressed all concerns from the reviewers.
Major revision
Question 1: It is Materials and Methods not Results
Response: Thanks for your valuable comments. We have revised the title, see line 111 .
Question 2: Indicate the full name of 'Pn, Cond, Ci', not just the abbreviation
Response: Thanks for your kind comments. We have revised the title, see line 363-369 .
Question 3: Is p<0.05 and not >
Response: Thanks for your kind comments. We have revised the contents, see the manuscript.
Question 4: The caption of figure 6 goes under the figure and not at the end of the discussions
Response: Thanks for your kind comments. We have revised the caption of figure 6, see page 21
Question 5: Throughout the document, check whether Bn and Ov are italicised. For figures: the quality of the images needs to be improved as some of them look grainy.
Response: Thanks for the reviewer's objective questions. We have revised figures in this manuscript (attachment of the figures).
As mentioned above, we tried our best to improve the manuscript and made some changes in the revised manuscript. These changes will not influence the content and framework of the manuscript.
We appreciate for reviewers and editors' warm work earnestly, and hope that the correction will meet with approval.
Reviewer 3 Report
The Authors of the article present an interesting paper because they investigated the glucose metabolism and nitrogen utilization in two Brassicaceae species in particular conditions, at specific concentration of bicarbonate and variable ammonium soil conditions. The topic of the paper is appealing because it not only investigates glucose metabolism and nitrogen assimilation in two plants such as Brassica napus (a non-karst-adaptable plant) and Orychophragmus violaceus (a karst-adaptable plant, but also because the information obtained provides interesting data given the ongoing climate change. The authors worked hard and the many results prove it. The paper is worthy of consideration, but in my opinion some aspects need to be reviewed. The introduction is adequate as references, but the text is often not very flowing, tiring to read, which could discourage the interested reader, but little familiar with the topic of the study. The results are very detailed, as well as the discussion part where figure 6 is present (usually there are no figures in the discussion section). Perhaps combining the two sections (results and discussion) would make the whole study more understandable and usable. Furthermore, the entire manuscript must be carefully checked so that each abbreviation is explained in full and at the first citation and is not present in the abstract, attention must be paid to punctuation, spaces, and when italics are necessary.
Please check:
What mean 3 between tables at page 15?
Line 489: 3.9 T?
At lines 494-495 “diverse letters in each value are significantly different by ANOVA (p > 0.05)”. It is better to rephrase the sentence “each value represents the mean ± SD (n=3), and different letters indicate significantly different values by Anova (p<0.05)”
At lines 552-553, please rephrase the sentence “Moreover, it was also attributed to superior PPP and RCRUBP of Ov plants”. Superior PPP and RCRUCP? It does not sound scientific.
Finally, despite the certificate of English language revision an additional check is necessary
Despite the certificate of English language revision, an additional check is necessary
Author Response
Comments: The Authors of the article present an interesting paper because they investigated the glucose metabolism and nitrogen utilization in two Brassicaceae species in particular conditions, at specific concentration of bicarbonate and variable ammonium soil conditions. The topic of the paper is appealing because it not only investigates glucose metabolism and nitrogen assimilation in two plants such as Brassica napus (a non-karst-adaptable plant) and Orychophragmus violaceus (a karst-adaptable plant, but also because the information obtained provides interesting data given the ongoing climate change. The authors worked hard and the many results prove it. The paper is worthy of consideration, but in my opinion some aspects need to be reviewed. The introduction is adequate as references, but the text is often not very flowing, tiring to read, which could discourage the interested reader, but little familiar with the topic of the study. The results are very detailed, as well as the discussion part where figure 6 is present (usually there are no figures in the discussion section). Perhaps combining the two sections (results and discussion) would make the whole study more understandable and usable. Furthermore, the entire manuscript must be carefully checked so that each abbreviation is explained in full and at the first citation and is not present in the abstract, attention must be paid to punctuation, spaces, and when italics are necessary.
Response: We gratefully thanks for the precious time the reviewer spent making constructive remarks. First of all, it needs to be understood by the reviewers that due to the many experiments set up (20 groups), we have developed a detailed discussion of the results. For ease of reading, first we list the maximum and minimum values so that they can be compared with other data. In addition, we have also listed the "significant" results in order to enhance the understanding of reviewers and readers. For example, section 3.6 (lines 434-439). Thus, most of the results are expressed in this way in our experiments. Of course, we believe that the reviewers' suggestions are very objective, and that you will eventually agree with our presentation and accept our changes to the main points below. Moreover, in the abstract, we presented the topic theme (lines 92-97) questions (lines 105-110), which we sincerely wish you accept for revision. Last but not least, the manuscript has also been double-checked, and the typos and grammar errors we found have been corrected (certificate see as attachment nt). In the following section, we summarize our responses to each comment from the reviewers. We sincerely hope that our responses have well addressed all concerns from the reviewers so as to our manuscript can be revised more comprehensively for publication.
Major revision
Question 1: What mean 3 between tables at page 15?
Response: Thanks for your valuable comments. We have revised the content, see page 15 .
Question 2: Line 489: 3.9 T?
Response: Thanks for your kind comments. We have revised the title of section 3.9.
Question 3: At lines 494-495 “diverse letters in each value are significantly different by ANOVA (p > 0.05)”. It is better to rephrase the sentence “each value represents the mean ± SD (n=3), and different letters indicate significantly different values by Anova (p<0.05)”.
Response: Thanks for the reviewer's objective questions. We have revised the contents of “p<0.05” in this article.
Question 4: At lines 552-553, please rephrase the sentence “Moreover, it was also attributed to superior PPP and RCRUBP of Ov plants”. Superior PPP and RCRUCP? It does not sound scientific.
Response: Thanks for your kind comments. We have revised the contents of line 574-575.
As mentioned above, we tried our best to improve the manuscript and made some changes in the revised manuscript. These changes will not influence the content and framework of the manuscript.
We appreciate for reviewers and editors' warm work earnestly, and hope that the correction will meet with approval.
Round 2
Reviewer 1 Report
Accepted. The authors responded well to my questions.